# Genetic deciphering of the antagonistic activities of the melanin-concentrating hormone and melanocortin pathways in skin pigmentation

Romain Madelaine[1], Keri J. Ngo[1,2], Gemini Skariah[1], Philippe Mourrain[1,3]*

**1** Department of Psychiatry and Behavioral Sciences, Stanford University, Stanford, California, United States of America, **2** Department of Developmental Biology, Stanford University, Stanford, California, United States of America, **3** INSERM 1024, Ecole Normale Supérieure, Paris, France

* mourrain@stanford.edu

**Data Availability Statement:** All relevant data are within the manuscript and its Supporting information files.

## Abstract

The genetic origin of human skin pigmentation remains an open question in biology. Several skin disorders and diseases originate from mutations in conserved pigmentation genes, including albinism, vitiligo, and melanoma. Teleosts possess the capacity to modify their pigmentation to adapt to their environmental background to avoid predators. This background adaptation occurs through melanosome aggregation (white background) or dispersion (black background) in melanocytes. These mechanisms are largely regulated by melanin-concentrating hormone (MCH) and α-melanocyte–stimulating hormone (α-MSH), two hypothalamic neuropeptides also involved in mammalian skin pigmentation. Despite evidence that the exogenous application of MCH peptides induces melanosome aggregation, it is not known if the MCH system is physiologically responsible for background adaptation. In zebrafish, we identify that MCH neurons target the pituitary gland-blood vessel portal and that endogenous MCH peptide expression regulates melanin concentration for background adaptation. We demonstrate that this effect is mediated by MCH receptor 2 (Mchr2) but not Mchr1a/b. *mchr2* knock-out fish cannot adapt to a white background, providing the first genetic demonstration that MCH signaling is physiologically required to control skin pigmentation. *mchr2* phenotype can be rescued in adult fish by knocking-out *pomc*, the gene coding for the precursor of α-MSH, demonstrating the relevance of the antagonistic activity between MCH and α-MSH in the control of melanosome organization. Interestingly, MCH receptor is also expressed in human melanocytes, thus a similar antagonistic activity regulating skin pigmentation may be conserved during evolution, and the dysregulation of these pathways is significant to our understanding of human skin disorders and cancers.

## Author summary

Melanocytes produce melanin, a natural skin pigment, for body coloration which helps to protect and camouflage an organism and to attract mates. Melanocytes are ubiquitous

**Funding:** This work was supported by grants (DK090065 and HL151576) from the National Institute of Health (https://www.nih.gov/) and the BrightFocus Foundation (https://www.brightfocus.org/). R.M was also supported by EMBO Long Term Fellowship (ALTF 413-2012). The funders had no role in study design, data collection and analysis, decision to publish, or preparation of the manuscript.

**Competing interests:** The authors have declared that no competing interests exist.

pigment cells in vertebrates and the genes underlying their development are well conserved, making fishes that possess the ability to modify their pigmentation, biologically relevant and successful models for human skin disorders. Many human skin diseases including albinism, vitiligo, and melanoma are derived from mutations in conserved pigmentation genes. However, much of the conserved molecular mechanisms behind these diseases and human pigmentation remain unknown. For instance, melanin concentrating hormone (MCH) was originally identified as a peptide that when injected, could make fish paler by promoting melanin aggregation but no mutants demonstrating an endogenous function for MCH in pigmentation have been reported. Here, we use zebrafish mutants of MCH and the MCH receptor to determine their specific genetic function in pigmentation. Additionally, we demonstrate that MCH has an antagonistic pigmentation function to the melanocortin system, where MCH expression promotes lighter pigmentation and melanocortin activity promotes darkening. Thus, we find that the balance between the MCH and melanocortin system activities are likely required for skin pigmentation and dysregulation of these pathways could underlie adverse human skin conditions.

## Introduction

Skin pigments play a role in the protection, attraction, and camouflage of an organism. Body coloration, such as pigment color and patterning, is derived from specialized pigment cells called melanocytes. Melanocytes produce melanin, a natural skin pigment, that is responsible for human coloring of the skin, eyes, and hair. In human skin, these cells offer protection against DNA damaging UV rays. Although some animals have additional pigment producing cells, melanocytes are the ubiquitous pigment synthesizing cell type in vertebrate animals [1]. Defects in melanocyte function can lead to skin pigmentation disorders, including vitiligo, albinism, and melanoma [1–3].

The conservation of melanocyte and melanophore-regulating genes across vertebrates has made fishes biologically relevant and successful models for uncovering the molecular mechanisms regulating melanocyte related human diseases [2–4]. For instance, zebrafish studies uncovered the first genetic evidence explaining human inter-ethnicity skin differences. Mutations in *SLC24A5*, a gene that encodes a cation exchanger and is associated with the *golden* zebrafish mutant, make the largest known contribution to skin color differences between people of European, Asian, and African ancestry [5, 6]. The light skin color variation may be regarded as a contributor to skin cancer susceptibility [5, 6].

Fish and mammals also share hypothalamic neuropeptides capable of changing skin pigmentation such as the melanin-concentrating hormone (MCH) and α-melanocyte–stimulating hormone (α-MSH) [7–11]. However, their genetic influence is not fully understood. MCH was originally identified in salmon *(Oncorhynchus keta)* as a peptide that can regulate skin pigmentation by promoting the aggregation of melanosomes [7, 8]. Indeed, injection of exogenous MCH into the bloodstream made salmons grow paler. However, no MCH fish mutant has yet been reported demonstrating an endogenous function in pigmentation. We previously discovered that fish genomes encode two different *MCH* genes, *Pmch1* and *Pmch2*, encoding MCH1 (17 amino acids) and MCH2 (19AA) respectively [12]. Whereas *Pmch1* and its corresponding MCH1 peptide resemble MCH originally found in salmon (sMCH), we found that the *Pmch2* gene and MCH2 peptide share genomic structure, synteny, and high peptide sequence homology with the mammalian *MCH* gene and peptide (mMCH) [12]. This finding

confirmed the conservation of this pathway across vertebrates. However, the specific genetic functions in pigmentation of MCH1/sMCH and MCH2/mMCH peptides remain unclear.

Where MCH expression promotes lighter pigmentation, the α-MSH/melanocortin system has the opposite effect in fish and mammals [13]. The melanocortin-1 receptor (MC1R) is a G-protein coupled receptor (GPCR) primarily located on melanocytes that regulate cAMP/protein kinase A (PKA) response through the binding of α-MSH encoded by the proopiomelanocortin *(POMC)* gene [14–16]. In mammals, MC1R signaling determines the amount of melanin pigment synthesized, affecting skin pigmentation [17]. Similarly, in fish and amphibians, α-MSH hormone peptides bind to MCRs to stimulate physiological body color changes [3, 18]. Contrary to MCH, α-MSH which is mainly expressed in the posterior part of the pituitary gland promotes the dispersion of melanosomes to regulate background adaptation [3, 18]. Interestingly, it has also been suggested that hypothalamic MCH peptide is released at the pituitary gland to inhibit the release of α-MSH [19, 20]. Because MCH decreases intracellular cAMP level while α-MSH increases it [3, 18], the balance between the activities of these two peptides is a key component of melanosome organization by microtubules and the actin cytoskeleton. Meanwhile, in human melanocytes this antagonistic activity is observed at the level of pigment production and not aggregation: α-MSH stimulates melanogenesis and MCH inhibits it [21]. These observations point to a conserved antagonistic relationship between α-MSH and MCH, whereby α-MSH promotes darker pigmentation and MCH promotes lighter pigmentation.

Still, genetic evidence demonstrating an endogenous role for the MCH pathway and its antagonistic interaction with α-MSH in vertebrate skin pigmentation are lacking. Here, we generated zebrafish mutants of the three MCH receptor genes (MCHR) as well as transgenic lines to control MCH peptides expression. We genetically demonstrate for the first time that the MCH system controls skin pigmentation in zebrafish. We revealed that both MCH1 and its paralogous peptide conserved in mammals, MCH2/mMCH, can regulate melanosome contraction. This effect is mediated through Mchr2 activity but not Mchr1a and 1b. *mchr2* and *pomc* double mutants also reveal epistatic interactions demonstrating *in vivo* that MCH and α-MSH have antagonizing roles in regulating background adaptation. Based on the conservation of MCH2/mMCH and α-MSH peptides and their receptors from fish to mammals, our findings uncover further genetic evidence of the antagonistic functions between the MCH and the melanocortin systems in vertebrate melanocyte homeostasis. It also suggests that their dysregulation could be part of the abnormal processes underlying human skin disorders such as vitiligo and melanoma.

## Results

### Mch axons projects to the pituitary gland

The zebrafish genome encodes three orthologues of MCH receptors (MCHR) [21]. Both *mchr1a* and *mchr1b* genes are broadly expressed in the brain, while mchr2 expression is specifically detected in skin melanocytes [12], suggesting a unique function for this receptor in skin pigmentation and background adaptation. However, the specific functions of the zebrafish MCH1 (similar to the salmon MCH) and MCH2 (similar to the mammalian MCH) peptides, and how the MCH signal can be transmitted from the brain to the MCHR expressed in the skin remained unclear.

We previously showed that the *Pmch1* and *Pmch2* genes are expressed in neurons located in the hypothalamus that project throughout the brain around the ventricles [12] to likely connect to MCHR1a and MCHR1b expressing neurons. In addition to the dense projections in the ventral telencephalon, thalamus, and posterior tuberal nucleus [12], we found the existence

of hypothalamic MCH fibers that extended toward the pituitary gland in the adult zebrafish brain (Fig 1A). However, as the adult pituitary is enclaved in a bone structure, it was lost in the brain tissue preparation and the actual connection of the MCH neurons to the pituitary was still unclear. Nevertheless, using an MCH antibody, we were able to show that the MCH peptide is present on the pituitary isolated from the adult zebrafish brain, revealing synaptic glomeruli-like structures (Fig 1B). Altogether, these observations show that hypothalamic MCH neurons can target the pituitary gland, suggesting a potential pathway via the blood stream to distribute the MCH peptide through the organism. To confirm the above hypothesis and reveal the entire MCH innervation on the pituitary gland, we used the zebrafish *Pmch2* gene promoter and we generated a stable *Tg(mch2:egfp)* reporter line (Fig 1C and S1 Fig). In the zebrafish *Tg(mch2:egfp)* larvae, we observed bright EGFP+ cell bodies in the lateral hypothalamus of both hemispheres that send converging projections to the ventral midline and densely innervated the pituitary gland. Further, mosaic EGFP expression in a single MCH neuron showed that axonal extensions innervate the pituitary at multiple points (S1(B) Fig). Consistent with these neuronal projections, using an MCH antibody [12], we showed similar projections and that the MCH peptides are preferentially accumulated on the pituitary gland, at the hypothalamo-neurohypophysis interface (Fig 1D, S1(D) and S1(E) Fig). This is reminiscent to other known pituitary peptides such as α-MSH and oxytocin (Fig 1E and 1F), where the α-MSH antibody stains cells in the posterior pituitary and the oxytocin antibody stains axons targeting the neurohypophysis [22]. This data validated the physical interaction between MCH neurons and the pituitary gland in zebrafish (Fig 1B–1F) as originally observed in salmon [8].

As the pituitary gland from fishes to mammals can be used as a portal to the body vasculature, this observation suggested a potential way in which MCH can control skin pigmentation of peripheral tissues via the bloodstream. To assess this possibility, we used an antibody for the vascular endothelial receptor marker, Kdrl, to visualize the surrounding vasculature architecture. We observed that the MCH peptides were nested in a ring-like vascular structure (Fig 1G and 1G'). The close proximity of MCH projections with blood vessels suggest that MCH peptides could indeed be released into the bloodstream via the pituitary to control a variety of physiological functions throughout the body.

## Mch1 and Mch2 overexpression reduce skin pigmentation

To investigate the role of both MCH1 and MCH2 peptides in the control of skin pigmentation, we generated two stable transgenic lines that express *mch1* and *mch2* after heat-shock; *Tg(hs:mch1)* and *Tg(hs:mch2)*. Because MCH peptide expression is known to increase when the fish is white adapted [12, 23], we raised these larvae on a black background to assess the activity of these peptides in the pigmentation process. After heat-shock, both MCH1 and MCH2 induced expression reduced melanosome dispersion even on a dark background (Fig 2A, 2B and 2C). We observed a reduction of 45% in skin pigment coverage after the induction of MCH expression compared to control larvae that do not carry the transgene but are also heat-shocked (Fig 2D). This result indicates that MCH2, like MCH1, can lead to paler skin when overexpressed and suggests that both peptides could bear redundant activity in the control of skin pigmentation and the physiological regulation of melanosome organization and background adaptation.

## MCHR2 is required for melanosome aggregation and background adaptation

We next sought to uncover the mechanism by which the MCH peptides induce changes in melanosome organization. MCH peptides act via MCH receptors, a type I G-protein coupled

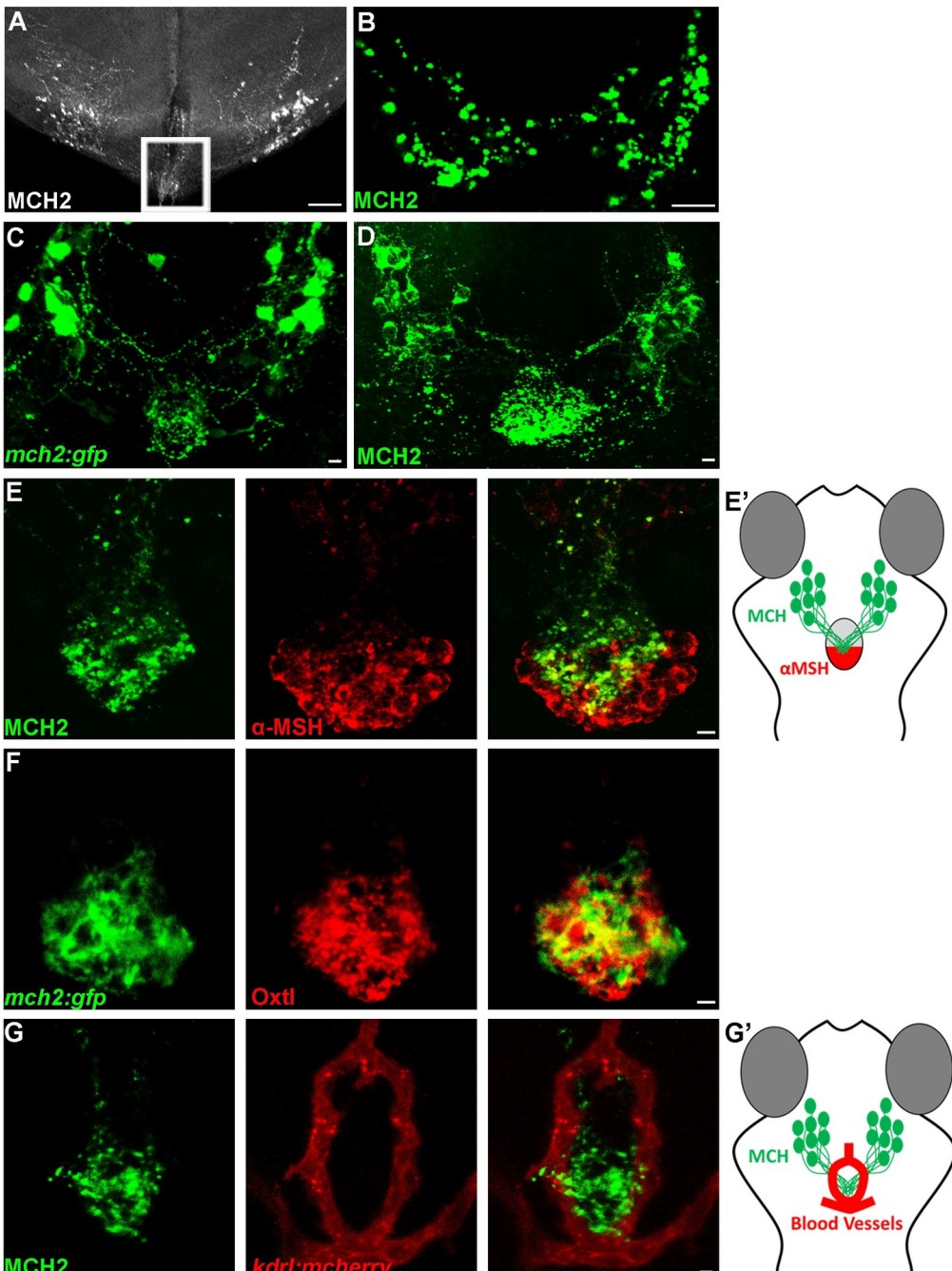

**Fig 1. The hypothalamic MCH2 peptide interacts with the neurohypophysis. (A-B)** Confocal projection of immunolabelling with MCH in the adult zebrafish brain (A) and pituitary gland (B). The outline area represents the hypothalamic MCH projection at immediate vicinity of the adult zebrafish pituitary, lost during the tissue preparation. **(C, D)** Confocal projection of immunolabelling with EGFP in *Tg(mch2:egfp)* hypothalamus (C) or with the MCH2 antibody (D) at 5 dpf, reveals that MCH2 neurons in the hypothalamus have axonal projections on the pituitary gland. **(E, E')** Confocal projection of double immunolabelling with MCH2 and α-MSH at 7 dpf, showing MCH cellular projections on the α-MSH+ domain (E). Schematic representation of the ventral part of the zebrafish brain showing the hypothalamic neurons expressing MCH2 targeting the posterior part of the pituitary gland expressing α-MSH (neurohypophysis) (E'). **(F)** Confocal projection of double immunolabelling with Oxtl and EGFP in *Tg(mch2:egfp)* hypothalamus at 7 dpf, demonstrating that MCH neurons target the neurohypophysis. **(G, G')** Confocal projection of double immunolabelling with MCH2 and blood vessels at 5 dpf, suggesting that MCH peptide can be released in the bloodstream (G). Schematic representation of the ventral part of the zebrafish brain showing the hypothalamic MCH2

terminal projections targeting the neurohypophysis surrounded by blood vessels (Hypothalamo-Hypophyseal system) (G'). Ventral (A-C) or (D-F) dorsal view with anterior up. Scale bars: 50 $\mu$m (A) or 10 $\mu$m (B-G).

receptor [19, 24]. Both *mchr1a* and *mchr1b* zebrafish genes are broadly expressed in the brain, while *mchr2* expression is specifically detected in the skin melanocytes from an early stage of larval development [12]. Humans have two MCH receptors, MCHR1 and MCHR2 [25], that are also largely expressed in the brain, but only MCHR1 expression has been shown in human melanocytes [10]. These observations indicate that MCHR expression in melanocytes is conserved across vertebrates, from human to zebrafish. While the MCHR1 receptor is known to be expressed in human skin and cultured melanocytes [10], suggesting a conservation of the function across vertebrates, a direct role of MCH receptors in controlling skin pigmentation has not previously been reported. To have a better understanding of the function of the MCH system in the regulation of skin pigmentation and background adaptation, we used a set of knock-out mutants for the three MCH receptors in zebrafish. We generated the *mchr1a* and *mchr2* mutants using the genome editing technique CRISPR/Cas-9, whereas the *mchr1b* mutant was generated by reverse genetics using TILLING. The *mchr1a* mutant harbors a small deletion of 4 bp, leading to a frameshift in the open reading frame (ORF) and the appearance of a premature stop codon (S2(A) Fig), while the *mchr1b* mutant carries a point mutation also leading to a premature stop codon (S2(B) Fig). To knock-out the *mchr2* gene, we generated a deletion of 4 nucleotides leading to a premature stop codon in the ORF (Fig 3A). To address the function of the MCH pathway in skin pigmentation we used a combination of these 3 MCH receptor mutants.

Through *in situ* hybridization, we previously showed that larval skin lacked *mchr1a* and *mchr1b* gene expression [12]. However, it remained possible that our detection method lacked the sensitivity required to truly capture the expression of these MCH receptors; thus, we observed the effects of knocking out *mchr1a* and *mchr1b* on larval skin to ensure that they were not involved in skin pigmentation. As expected, both *mchr1a* and *mchr1b* single homozygous mutants did not show any defects in skin pigmentation or background adaptation. To assess if there was partial redundant expression and function of these two receptors that was causing a normal phenotype in the single knock-out mutants, we crossed single mutants to generate a double mutant for *mchr1a* and *mchr1b*. The double mutant also showed normal skin pigmentation and adaptation to a white background (S2(C), S2(D) and S2(E) Fig),

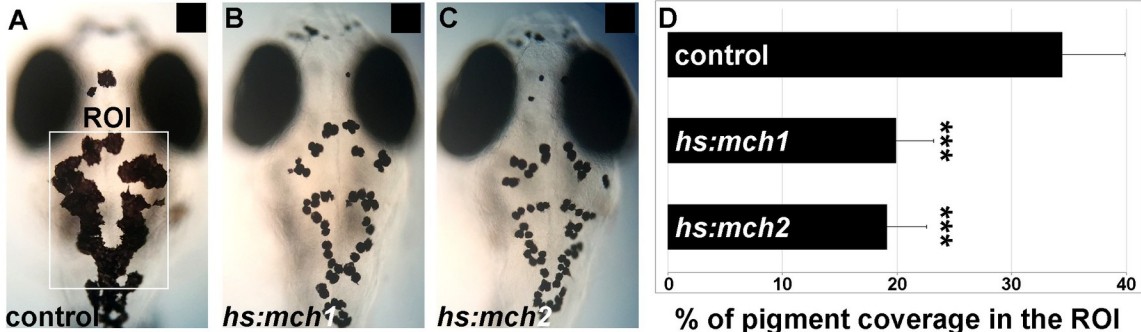

**Fig 2. Mammalian orthologous MCH2 peptide induces melanosome contraction. (A-C)** Dorsal melanocytes of control (A), *Tg(hs:mch1)* (B) or *Tg(hs:mch2)* (C) black adapted larvae at 7 dpf following heat-shock. **(D)** Melanosome coverage was quantified in control, *Tg(hs:mch1)* or *Tg(hs:mch2)* in region of interest (ROI). The ROI has been defined as an area from the posterior of the eyes to the posterior of the hindbrain. Error bars represent s.d. *P<0.05, **P<0.001, ***P<0.0005, determined by t-test, two-tailed.

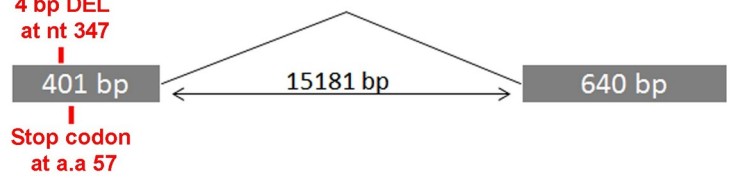

**A Gene editing at the *mchr2* locus:**

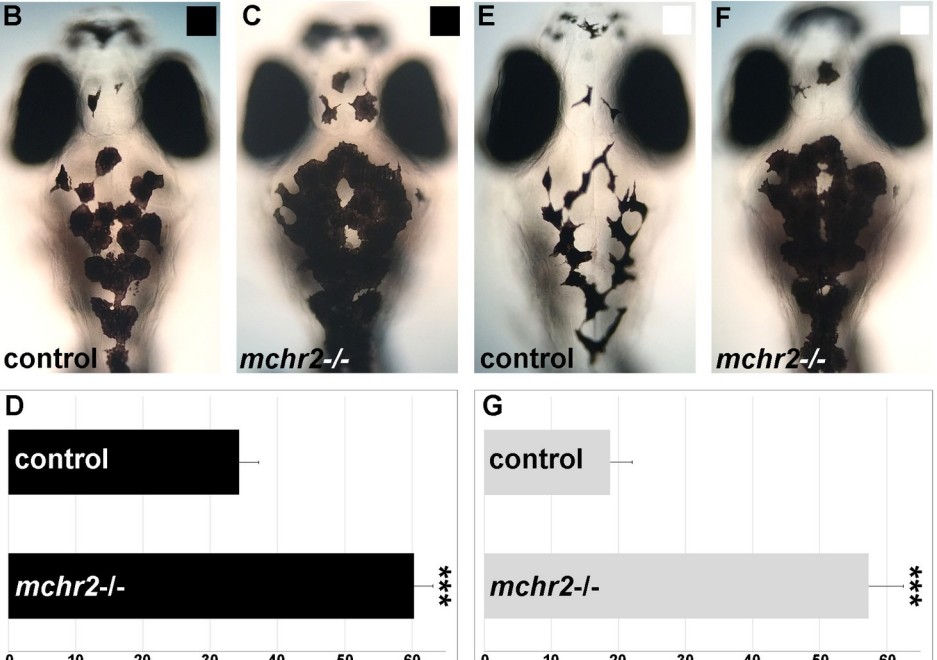

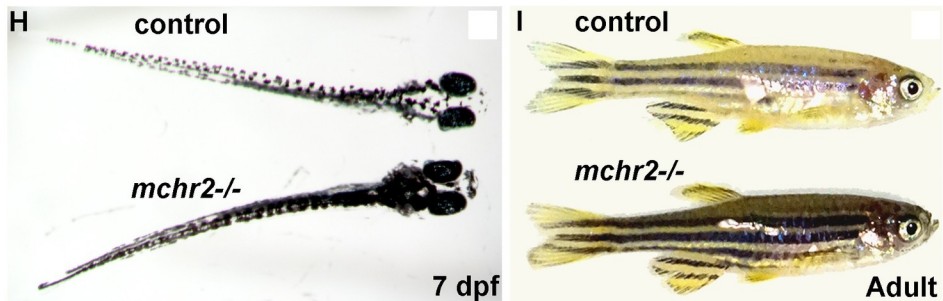

**Fig 3. Loss of Mchr2-dependent signaling leads to melanosome dispersion.** **(A)** Schematic representation of the *mchr2* locus and the small genomic deletion induced by CRISPR/Cas-9 leading to a premature stop codon in the mchr2 coding sequence. **(B, C)** Dorsal melanocytes of control (B) or *mchr2* homozygous mutant (C) black adapted larvae at 7 dpf. **(D)** Melanosome coverage was quantified in control or *mchr2* homozygous mutant black adapted larvae at 7 dpf in ROI, an area from the posterior of the eyes to the posterior of the hindbrain. **(E, F)** Dorsal melanocytes of control (E) or *mchr2* homozygous mutant (F) white adapted larvae at 7 dpf. **(G)** Melanosome coverage was quantified in control or *mchr2* homozygous mutant white adapted larvae at 7 dpf. **(H, I)** Larval and adult *mchr2* homozygous mutant phenotypes are characterized by melanosome dispersion on a white background, indicating that MCH signaling is required to promote melanosome contraction in melanocytes. Dorsal view with anterior up (B, C, E, F and H). Lateral view (I). Error bars represent s.d. $^*P < 0.05$, $^{**}P < 0.001$, $^{***}P < 0.0005$, determined by t-test, two-tailed.

suggesting that MCH signaling in melanocytes is still functional in this genetic context. While we cannot totally exclude that one of these *mchr1* mutants is not a null allele despite the presence of premature stop codons in their ORFs, these results strongly suggest that *mchr1a* and *mchr1b* are not involved in the control of melanosome concentration and skin pigmentation.

After excluding a role for *mchr1a* and *mchr1b* in the regulation of skin pigmentation and melanosome organization, we hypothesized that due to the restricted expression of MCHR2 in melanocytes [12], and its sequence conservation with the human MCHR1 expressed in the skin (S2(F) Fig), MCHR2 may be the functional effector of the MCH pathway in the control of skin pigmentation. Strikingly, the *mchr2* mutant shows an increase of 75% in skin pigment coverage when raised on a black background (Fig 3B, 3C and 3D). In this mutant, melanocytes harbor an extensive dispersion of the melanosomes in the cell, indicating a potent inactivation of MCH signaling. Additionally, the *mchr2* mutant is not able to adapt to a white background as larvae (Fig 3E–3H; 200% increase in pigment coverage in the mutant) and as adults (Fig 3I), revealing that MCH signaling is a critical component of the background adaptation system throughout life. Finally, *mch1* or *mch2* overexpression in the *mchr2* mutant background failed to rescue the pigmentation phenotype or reduce melanin dispersion (Fig 4A–4E). This result demonstrates that MCH peptides need functional MCHR2 in melanocytes to promote melanin concentration. In addition, it supports the observation that MCHR1a and MCHR1b do not have compensatory functions in skin pigmentation control. Altogether, these data provide the first genetic evidence that demonstrate the direct physiological role of MCH receptors in the control of melanosome organization in vertebrates.

## Antagonistic activity between MCH and α-MSH controls skin pigmentation

In the *mchr2* mutant where MCH signaling is abolished in melanocytes, we observed an increase in skin pigment coverage. However, *mchr2* mutants show normal *tyrosinase* expression levels in melanocytes (S3(A) and S3(B) Fig), suggesting that these cells develop normally in this mutant. It also suggests that the pigmentation phenotype is not due to an increase in melanin synthesis. In addition, it has been previously shown that α-MSH induces melanosome dispersion [3, 18, 26] and in the melanocortin-1 receptor *(mc1r)* knock-down, there is inhibited melanosome dispersion in zebrafish [22]. Altogether, these observations led us to hypothesize that because MCH signaling is not functional in *mchr2-/-* melanocytes, the melanocortin pathway (POMC/α-MSH-MC1R) is responsible for inducing melanin dispersion.

We next asked how α-MSH, cleaved from the proopiomelanocortin precursor (POMC), may be involved in the skin pigmentation defects observed in the *mchr2* mutant. Teleosts possess two POMC paralogs, *pomca* and *pomcb*, originating from an ancient whole-genome duplication event [27]. During teleost evolution, the expression domains of the *pomc* paralogs underwent a process of sub-functionalization, partitioning *pomca* expression to the pituitary gland and the ventral hypothalamus while *pomcb* expression was partitioned to the brain preoptic area [27–30]. Based on these different expression patterns, we chose to focus on *pomca*, reasoning that the *pomca* paralog was more likely to be the α-MSH precursor within the pituitary. In the *mchr2* knock-out, we did not detect any variation in the expression of *pomca* mRNA or α-MSH small peptides (S3(C)–S3(F) Fig), suggesting that the melanocortin pathway could be responsible for the skin pigmentation phenotype. Supporting this hypothesis, α-MSH expression promotes melanosomes dispersion [3, 18], a phenotype reminiscent to the *mchr2* mutant, indicating that POMC/α-MSH activity may be responsible for the melanosome organization and background adaptation defects observed in the *mchr2* mutant.

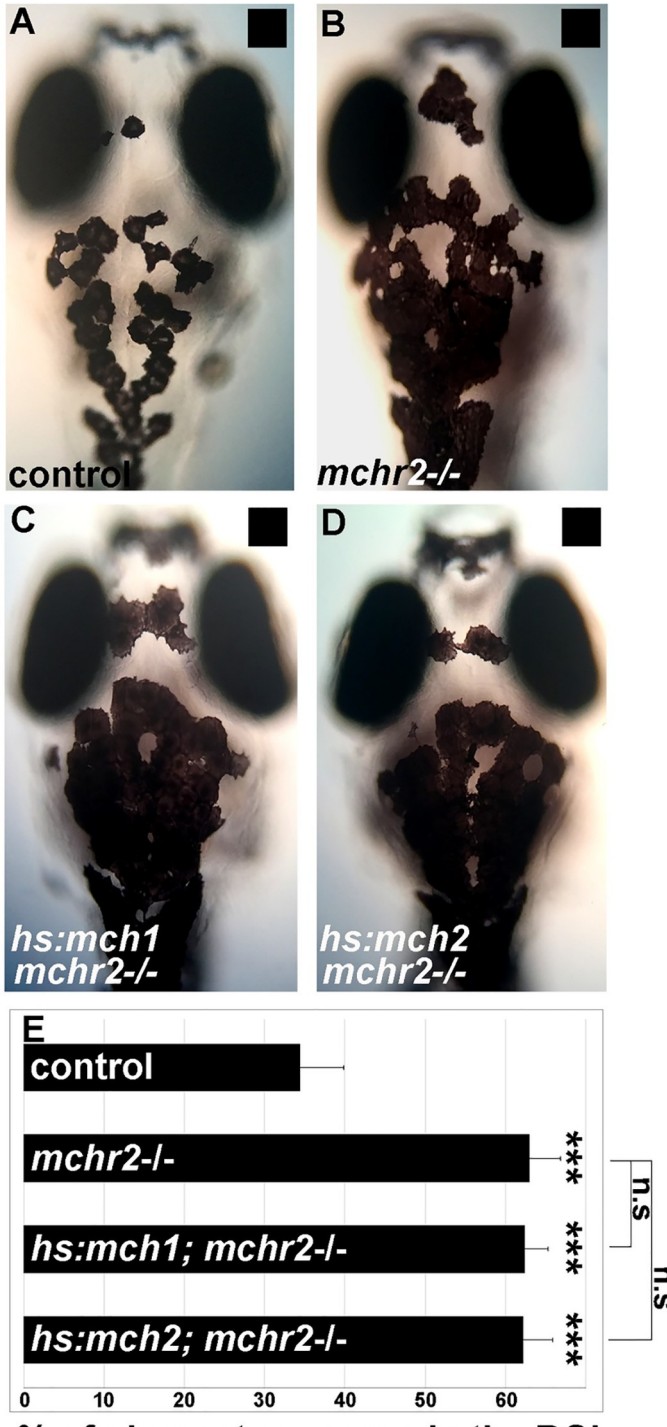

**Fig 4. MCH-dependent melanosome contraction relies on Mchr2 activity. (A-D)** Dorsal melanocytes of control (A), *mchr2* homozygous mutant (B), *mchr2* homozygous mutant; *Tg(hs:mch1)* (C) or *mchr2* homozygous mutant; *Tg(hs:mch2)* (D) black adapted larvae at 7 dpf following heat-shock. **(E)** Melanosome coverage was quantified in control, *mchr2* homozygous mutant, *mchr2* homozygous mutant; *Tg(hs:mch1)* or *mchr2* homozygous mutant; *Tg(hs:mch2)* larvae at 7 dpf. Dorsal view with anterior up. Error bars represent s.d. $^*$P<0.05, $^{**}$P<0.001, $^{***}$P<0.0005, determined by t-test, two-tailed.

To investigate the role of *pomc* in the pigmentation process in the *mchr2* mutant, we generated a knock-out of the *pomca* gene (Fig 5A–5C'), the only precursor of α-MSH which is expressed in the pituitary gland (S3(G) and S3(H) Fig). The mutant allele carries two deletions generated by CRISPR/Cas9 editing (one in each coding exon of the *pomca* ORF; Fig 5A). Double immunolabeling with α-MSH and MCH antibodies confirms that the *pomca* homozygous mutant lacks α-MSH expression but still maintains MCH expression when compared to control sibling larvae, suggesting impaired melanocortin signaling (Fig 5B–5C'). Consistent with the previously described phenotype of *mc1r* depletion [22], *pomca* mutant larvae are not able to adapt to a black background (Fig 5D and 5E), as shown by a reduced skin pigmentation coverage of 50% compared to control siblings (Fig 5F). Moreover, reported *mc1r* mutants have disrupted dorso-ventral countershading. Where fish normally have a dark dorsum and a light ventrum, the *mc1r* mutants have hyperpigmented ventrums and less dorsal melanophores [31, 32]. Unlike these *mc1r* knock-outs, the *pomca* knock-outs have relatively normal countershading as they display dark dorsums (S4(A)–S4(D) Fig) and white ventrums (S4(E) and S4(F) Fig). Differences in the number of dorsal melanophores and xanthophores were found to be statistically insignificant (S4(G) Fig), indicating that the observed pigmentation changes are indeed predominantly due to background adaptation.

The skin pigmentation phenotype of the *pomca-/-* mutant is reminiscent to the pigment phenotype when MCH is overexpressed (Figs 2A–2D and 5D–5F), indicating that the loss of α-MSH function leads to a predominant activity of the MCH system. Interestingly, the knock-out of *pomca* in the *mchr2* mutant does not rescue the *mchr2* mutant phenotype as double *mchr2-/-, pomca-/-* mutant larvae are not able to adapt to a white background and they maintain extensive melanosome dispersion (Fig 5G–5J) more akin to a black adapted phenotype (Fig 5D). This result indicates that additional factors other than α-MSH can promote melanosome dispersion in the *mchr2* mutant genetic background in larvae.

Surprisingly, in juvenile and adult fish, the *mchr2-/-, pomca-/-* double mutant phenotype appears to be different than in larvae. Indeed, starting at about 1-month post fertilization, we observe that skin pigmentation in the *mchr2-/-, pomca-/-* double mutant is no longer identical to the *mchr2* mutant, and becomes reminiscent to the *pomca-/-* phenotype in adults (Fig 6A–6D). Strikingly, adult double mutants are not able to adapt to a black background and harbor a pattern of pigmentation similar to *pomca* mutant fish where striped patterning is lighter than the stripes of *mchr2* mutants (Fig 6E–6H). These observations indicate that the inability of the adult *mchr2* mutant to adapt to a white background is due to the predominant activity of the melanocortin pathway, and that abolishing this signaling is sufficient to promote melanosome contraction in the *mchr2* homozygous mutant. Altogether, these data bring functional evidence to demonstrate that the antagonistic activity between MCH and melanocortin signaling is a major regulator of melanosome organization, skin pigmentation, and reveals a different mode of interaction between these two pathways during development and adult life.

## Discussion

This study genetically demonstrates that the MCH and the α-MSH/melanocortin systems interact to control skin pigmentation in zebrafish and likely all vertebrates. Through fluorescent labeling studies, we show a tight spatial association of MCH peptides with the pituitary blood portal, suggesting neuropeptide release into the peripheral circulation whereby body wide responses can be induced. We and others have demonstrated that both MCH1 and MCH2 expression increase when the fish is white adapted, indicating a physiological role in the control of background adaptation for these peptides [12, 23]. Here we show that much like MCH1, overexpression of MCH2 can reduce skin pigmentation in zebrafish, raising the

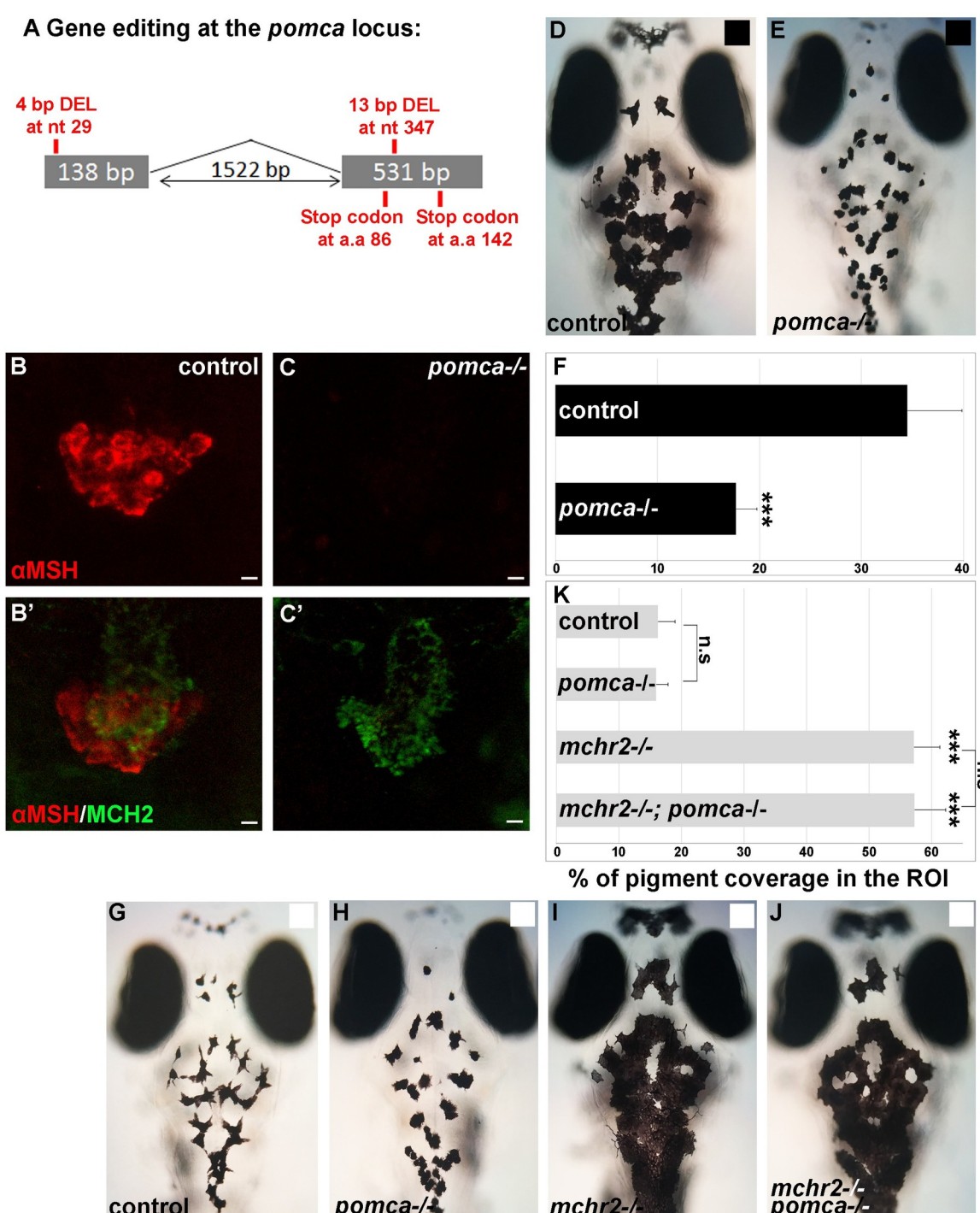

**Fig 5. *pomca* loss of function leads to melanosomes contraction, but is not sufficient to rescue the *mchr2* mutant phenotype in larvae. (A)** Schematic representation of the *pomca* locus and the small genomic deletions induced by CRISPR/Cas-9 leading to premature stop codons in the *pomca* coding sequence. **(B-C')** Confocal projection (B, C) or section (B', C') of double immunolabelling with MCH2 and α-MSH antibody at 5 dpf, showing that MCH expression, but not α-MSH, is detected in the *pomca* homozygous mutant. **(D, E)** Dorsal melanocytes of control (D) or *pomca* homozygous mutant (E) black adapted larvae at 7 dpf. **(F)** Melanosome coverage was quantified in control or pomca homozygous mutant black adapted larvae at 7 dpf. **(G-J)** Dorsal melanocytes of control (G), *pomca* homozygous mutant (H), *mchr2* homozygous mutant (H) or *pomca; mchr2* double homozygous mutant (J) white adapted larvae at 7 dpf. **(K)** Melanosome coverage was quantified in control, *pomca* homozygous mutant, *mchr2* homozygous mutant or *pomca; mchr2* double homozygous mutant white adapted larvae at 7 dpf. Dorsal view with anterior up (D, E and G-J). Scale bars: 10 μm. Error bars represent s.d. *P<0.05, **P<0.001, ***P<0.0005, determined by t-test, two-tailed.

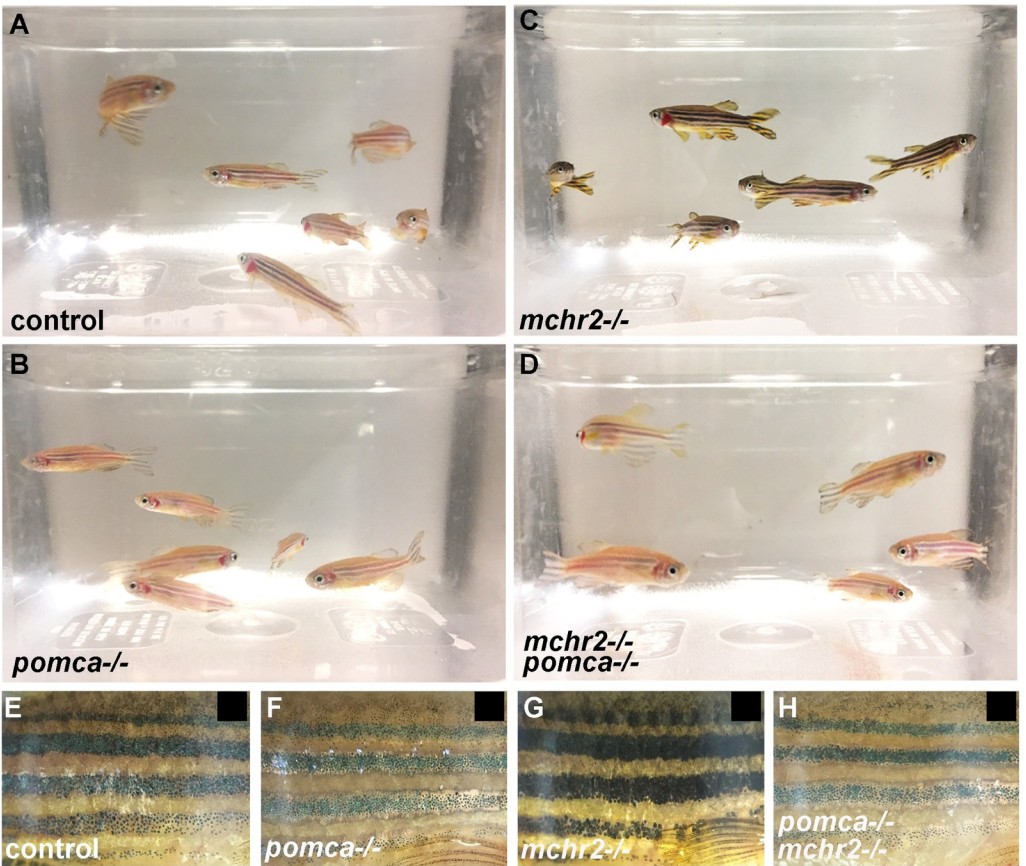

**Fig 6. MCH signaling counteracts melanocortin system activity to promote melanosome contraction in adult zebrafish. (A-D)** Loss of function phenotype on skin pigmentation from white adapted adult control **(A)**, *pomca* **(B)**, *mchr2* **(C)** or double *mchr2; pomca* **(D)** homozygous mutant. **(E-G)** Loss of function phenotype on skin pigmentation from black adapted adult control **(E)**, *pomca* **(F)**, *mchr2* **(G)** or double *mchr2; pomca* **(H)** homozygous mutant. Similar to the *pomca* homozygous mutant, *pomca; mchr2* double homozygous mutant phenotype is characterized by melanosome contraction on a black background, indicating that *pomca* loss of function is sufficient to allow melanosome contraction in absence of MCH signaling in adult zebrafish. Lateral view (E-G).

possibility that the same gene in mammals could also affect pigmentation. In mammals, due to redundant pigmentation systems, it is possible that the disrupted gene is not sufficient to cause a skin disorder but instead can worsen pathology and the outcome of the disease. Additionally, of the MCH receptor mutants, only the *mchr2-/-* knockout displayed a changed pigment phenotype. Thus, released MCH peptides require functional MCHR2 but not MCHR1a or MCHR1b to affect melanosome organization, indicating that MCHR2 can relay signals from both MCH peptides. Contrary to zebrafish, MCHR2 expression in humans was not detected in skin tissues or melanocytes [10]. Since zebrafish MCHR2 is homologous to human MCHR1 and MCHR2, it is likely that despite bearing the same signal transduction, over the course of evolution from teleosts to mammals, the expression patterns of the MCH receptors were swapped.

Consistent with the idea that the MCH and the α-MSH/melanocortin systems act antagonistically to each other, the *pomca-/-* larval phenotype resembles the MCH1 and MCH2 overexpression phenotype where skin pigmentation is reduced. The *pomca-/-* mutant remains pale throughout its life; whereas, the *mchr2-/-* retains dark pigmentation, and both mutants lose

their ability for background adaptation, indicating that both the melanocortin and MCH systems are required for the oscillation between melanosome dispersion and contraction. Several human skin pigmentation disorders and diseases originate from irregularities in melanocyte development and function. However, the causal relationships between the MCH and α-MSH regulatory systems and prevalent skin diseases such as melanoma and vitiligo are still poorly understood.

Melanoma is the most lethal skin disease originating from abnormalities in melanocyte development. The α-MSH receptor, MC1R, is expressed in skin cells like melanocytes [33], and variants in MC1R are known risk factors for melanoma [17]. Indeed, previous clinical studies indicate that there is a correlation between increased α-MSH levels and melanoma progression [34, 35]. Although no direct role for MCH in melanoma has been identified, MCHR1 has been found to be expressed in human melanocytes and melanoma cells [36], potentially allowing for MCH signaling to occur. The highly conserved interactions between the MCH and melanocortin pathways in controlling the development and function of melanocytes introduces the possibility that MCH variants might also be risk factors for melanoma, whereby changes in MCH affect POMC/α-MSH signaling.

In humans, vitiligo is a skin depigmentation disease caused by the loss of melanocytes [37]. The causes of the disease are not fully understood, but because vitiligo patients present autoantibodies and autoreactive T lymphocytes against melanocyte antigens, an autoimmune theory has been proposed [36]. In some cases, autoantibodies against MCHR1 were found in the serum of vitiligo patients and it has been hypothesized that they can modulate receptor activity [38, 39]. Indeed, MCHR1 autoantibodies have been shown to block the binding of MCH peptide with MCHR1 in a competitive manner and/or inhibit the activation of the receptor by blocking the release of intracellular calcium in response to MCH [39]. The authors hypothesized that the blocking of MCHR1 activity may affect normal melanocyte behavior via the lack of α-MSH downregulation. In mammals, α-MSH is known to positively regulate melanin synthesis [37] and because increase in melanin synthesis can lead to melanocyte death [37, 40, 41], the maintenance of a physiological balance between MCH and POMC/α-MSH signaling is likely critical to the skin pigmentation process. Altogether, these observations and our study, support the notion that the antagonistic activity between MCH and the melanocortin system is a master regulator of the melanosome organization conserved across vertebrates.

Surprisingly though, we observed that abolishing the opposing functions of MCH and α-MSH does not lead to the same phenotype in larvae and adults, revealing that their antagonistic activity is not integrated in the same manner during development. The *pomca/mchr2* double knock-out mutant lacks the ability for background adaptation and has dark pigmentation in the larval stage akin to the *mchr2-/-* single mutant phenotype. However, the pigmentation of the *mchr2/pomca* double mutant becomes paler in adulthood, resembling the *pomca-/-* single mutant. One possible explanation for this discrepancy is the redundant activity with other molecular factors that are expressed differentially during life, like sexual maturation hormones or other neuropeptidergic signals. For example, in young fish, it is possible that additional uncovered neuropeptides control melanosome organization and have redundant activity with the melanocortin system, thus preventing the rescue of the *mchr2-/-* phenotype via the loss of *pomca* function. However, in adults, our results indicate that the *pomca* knock-out phenotype is dominant over the *mchr2* mutant phenotype, suggesting that additional factors besides MCH can control melanosome contraction.

Such factors could include other melanocortin receptor antagonists like agouti signaling protein (Asip) and agouti-related protein (Agrp) or other signaling pathways. It was reported that mutations in galanin signaling downregulates peripheral levels of thyroid hormones, leading to fewer stripes and paler appearance [42]. Importantly, galanin and MCH have both been

involved in sleep and pigmentation regulation further suggesting common physiological functions and targets [42, 43]. Conceivably, combined pigmentation effects in multiple signaling pathways like galanin and melanocortin signaling could cause different phenotypes during development. In mammals, *asip* has been extensively reported to be expressed in dermal papillae cells and to be involved in pigmentation by competing with α-MSH for MC1R and MC4R binding and causing subsequent changes in melanin composition [44–48]. It has also been reported that *asip* can affect melanocyte differentiation and maturation in mice [49]. Meanwhile, mammalian *agrp* is mainly expressed in the hypothalamus where it has predominantly been explored in the context of regulating energy balance [50, 51]. In contrast to tetrapods, teleosts have four endogenous antagonists: *asip1, asip2, agrp1* and *agrp2* due to genome duplication [52, 53]. Previous studies in fish have shown that *asip1, agrp1*, and *agrp2* are expressed in the skin and brain [54–56], where *asip1* and *agrp2* are expressed in the pineal gland and *agrp1* is expressed in the hypothalamus, making them promising candidate factors for the regulation of melanosome organization in background adaptation.

Supporting the previous observation, injection of *asip1* mRNA induces dorsal skin paling in multiple fish species and can reduce tyrosinase-related protein-1 *(tyrp1)* expression, a key enzyme in melanin synthesis [52, 57]. Moreover, in medaka *(Oryzias latipes)*, *asip1* has been found to inhibit α-MSH-induced melanosome movement via Mc1r binding competition [55]. *Agrp2* also antagonizes Mc1r and previous demonstration has shown that *agrp2* is required for melanin concentration [23], raising the possibility that *asip1* and/or *agrp2* may be able to promote melanosome concentration in the double *mchr2/pomca* mutant. In this case, our observations also suggest that Asip1 or Agrp2 activity alone is not sufficient to induce melanosome contraction and to antagonize the melanocortin signaling in the *mchr2* mutant; either because it acts partially through Mchr2 signaling likely via *mch1* and *mch2* upregulation [23]; or because MCH and Asip1/Agrp2 activity need to synergize to counteract the melanocortin pathway. The function of Asip and Agrp peptides remains to be examined in the context of the antagonistic actions of the MCH and α-MSH/melanocortin systems in melanosome organization, where they may have a conserved role in vertebrate pigmentation and in the etiology of skin disorders.

Due to frequent dimerization among GPCRs [58, 59], we cannot exclude the possibility that heterodimerization of the MCR subtypes contribute to the observed discrepancy in skin phenotype. While there is no direct evidence that there is cross-binding between MCH and the melanocortin ligands with their receptor subtypes or that the ligands competitively inhibit each other's binding to their respective receptors [13], some studies suggest that MCH may have some affinity for MC5R since high concentrations of MCH can mimic the effects of α-MSH in tissues and cell cultures expressing MC5R [60]. Indeed, there have been previous reports that MCH exhibits self-antagonistic, MSH-like activity at high concentrations in teleost melanophores whereby pigment dispersion is induced [61, 62] via an unknown mechanism. Although pigmentation is not mainly attributed to MC5R activity as it is with MC1R, MC5R is weakly expressed in the brain and is widely distributed in more peripheral regions like in the skin [63, 64] and is able to heterodimerize to MC1R [65]. It has been reported that MC1R/MC5R heterodimerizations weakens α-MSH signaling by inhibiting cAMP accumulation and that this ligand-selective receptor complex may be able to influence color changes in teleosts [65]. Thus, reduced α-MSH signaling and the self-antagonistic MCH pigment activity could be explained by competitive binding of MCH and α-MSH for the MC1R/MC5R heterodimer and this mechanism is contributing to the different phenotypes observed in the *pomca/mchr2* double knock-outs throughout development. Additionally, studies have shown that mammalian MCRs can dimerize [66–68], leading to the possibility that MC1R/MC5R dimers appear

in amniotic vertebrates and can similarly affect pigmentation through balancing MCH and α-MSH activities.

As skin pigmentation regulation is critical for many aspects of animal life including UV protection, camouflage, mimicry, aposematism (warning coloration) and sexual selection/reproduction fitness, to name a few, its precise regulation very likely requires a complex interplay of signals from the brain and body. Commonalities across vertebrates and amenable fish genetics continues to reveal the depth of these conserved mechanisms.

## Materials and methods

### Fish lines and developmental conditions

**Ethics statement.** Embryos were raised and staged according to standard protocols [69]. Our IACUC animal protocol (#9935) is approved by the American Association for the Accreditation of Laboratory Animal Care (AAALAC) in accordance with Stanford University animal care guidelines.

*Tg(kdrl:ras-mCherry)* [70] was used to visualize blood vessels. *mchr1b* mutant was generated by the zebrafish mutation project (Sanger Institute) and obtained from the ZIRC. Heat-shocks were performed in a water bath at 37°C for 1 hour. Embryos were fixed overnight at 4°C in 4% paraformaldehyde/1xPBS, after which they were dehydrated through an ethanol series and stored at −20°C until use.

### CRISPR/Cas-9 generation of zebrafish mutants

To generate *mchr1a, mchr2* and *pomca* mutants, we took advantage of the previously described genome editing method [71]. We used gRNA targeting the ORF (*mchr1a*: 5'-GTATGTGGC TACTGTCCACC-3', *mchr2*: 5'-CAATCTACGGCGTCCTGTGC-3' and *pomca*: ex1-5'-GAG GATGTTGTGTCCTGCT-3' and ex2-5'-GTGGGGCAAACCGGTCGG-3'). Co-injection of these gRNA with the *Cas9* mRNA resulted in small deletions. F0 carriers for the deletion in the germline were identified by sequencing of the alleles on the clutches and outcrossed to obtain F1 heterozygous mutants. F2 mutants homozygous for the deletion were also identified by sequencing. Our *pomca* knock-out carries the homozygous mutant allele containing both deletions.

### Plasmid construction and transgenic line establishment

For the generation of *Tg(mch2:egfp)*, 5kb of the zebrafish *mch2* promoter was amplified by PCR on genomic DNA and was cloned in the p5E 5' entry vector of the tol2kit. For the generation of heat-shock lines, *mch1* and *mch2* ORF were amplified by PCR from zebrafish cDNA and cloned in the pME middle entry vector. The appropriate entry and middle entry clones were mixed with the SV40pA 3' entry vector and recombined into the Tol2 transposon destination vector. To establish stable transgenic lines, plasmids were injected into one-cell stage embryos with the Tol2 mRNA transposase [72].

### *In situ* hybridization and immunostaining

*In situ* hybridizations were performed as previously described [73]. *tyrosinase (tyr) pomca* and *pomcb* ORF was cloned in a pCS2+ vector using zebrafish cDNA and antisense DIG labelled probes were transcribed using the linearized pCS2+ plasmid containing the ORF. *In situs* were revealed using either BCIP and NBT (Roche) or Fast Red (Roche) as substrates. Immunohistochemical stainings were performed as previously described [74], using either anti-GFP (1/1000, Torrey Pines Biolabs), anti-α-MSH (1/10000, Millipore), anti-OXT (1/500, Bachem) or

anti-MCH (1/500, Phoenix Pharmaceuticals) as primary antibodies and Alexa 488 or Alexa 555-conjugated as secondary antibodies (1/1000, Molecular Probes).

## Image acquisition and quantification of pigmentation coverage

Confocal acquisitions were carried out using a Leica SP5 confocal microscope (Stanford Cell Science Imaging Facility) and melanosomes were imaged using light microscopy. Images were prepared using Photoshop software (Adobe). For quantification of pigment coverage, images were analyzed using ImageJ software. Statistical analyses associated with each figure are reported in the figure legends.

## Melanophore and xanthophore counts

Following previously reported protocols in adult fish [31, 32], melanophores and xantho-phores cells were counted in 1 $mm^2$ regions in the dorsal area from the edge of the head to the dorsal fin. To contract melanosomes, fish were anesthetized in tricaine and immersed in 10 mg/ml epinephrine (Sigma) solution for 30 min. Images were analyzed using ImageJ software. Data are expressed as mean ± standard error of the mean (SEM) and evaluated by t-test, where a p-value<0.05 was determined to be statistically significant. n = 2 fish per category.

## Supporting information

**S1 Fig. *mch2* reporter line demonstrates that Mch axons reach the pituitary gland. (A, B)** Confocal projection of immunolabelling with EGFP in *Tg(mch2:egfp)* hypothalamus at 5 (A) or 3 (B) dpf, reveals that MCH2 neurons in the hypothalamus have axonal projections on the pituitary gland. **(C)** Confocal section of double immunolabelling showing overlap in the expression of endogenous MCH2 peptides and EGFP in *Tg(mch2:egfp)* at 5 dpf. **(D, E)** Confocal projection of immunolabelling with EGFP in *Tg(mch2:egfp)* (D) or with MCH2 (E) at 7 dpf, showing identical projection patterning of MCH2 neurons on the neurohypophysis with both tools. Ventral view with anterior up. Scale bars: 10 $\mu$m.
(TIF)

**S2 Fig. *mchr1a* and *mchr1b* are not involved in melanosome organization. (A)** Schematic representation of the *mchr1a* locus and the small genomic deletion induced by CRISPR/Cas-9 leading to a premature stop codon in the mchr1a coding sequence. **(B)** Schematic representation of the *mchr1b* locus and the nucleotide substitution leading to a premature stop codon in the mchr1b coding sequence. **(C, D)** Dorsal melanocytes of control (C) or *mchr1a; mchr1b* double homozygous mutant (D) white adapted larvae at 7 dpf. **(E)** Melanosome coverage was quantified in control or *mchr1a; mchr1b* double homozygous mutant white adapted larvae at 7 dpf. Dorsal view with anterior up. **(F)** Amino acids sequence alignment between the zebrafish Mchr2 and the human MCHR1 proteins. The highlighted sequence shows the conservation of the G-protein receptor domain. Dorsal view with anterior up. Error bars represent s.d. *P<0.05, **P<0.001, ***P<0.0005, determined by t-test, two-tailed.
(TIF)

**S3 Fig. *pomca* expression in the posterior pituitary gland. (A, B)** Whole-mount *in situ* hybridization against *tyrosinase (tyr)* in larvae at 3 dpf, showing that melanocyte organization appears to be normal in the *mchr2* homozygous mutant. **(C, D)** Whole-mount *in situ* hybridization against *pomca* in larvae at 5 dpf, showing that *pomca* expression appears to be normal in the *mchr2* homozygous mutant. **(E, F)** Confocal projection of immunolabelling with α-MSH, showing that α-MSH expression appears to be normal in the *mchr2* homozygous mutant. **(G, H)** Confocal section of double *in situ*/immunolabelling with *pomca* (G) or *pomcb*

(H) in *Tg(mch2:egfp)* at 5 dpf, showing that only *pomca* is expressed in the pituitary gland. Dorsal view with anterior up (A, B). Ventral view with anterior up (C-F, I and J). Scale bars: 100 $\mu$m (A-D) or 10 $\mu$m (E, F, I and G).
(TIF)

**S4 Fig. *pomca* mutant displays normal countershading. (A-D)** Dark dorsums of 3-year-old *pomca-/-* and WT control sibling fish before (A, C) and after (B, D) epinephrine treatment to contract melanosomes. **(E, F)** Light ventrums of adult control (E) and *pomca-/-* (F) fish. **(G)** Quantification of dorsal melanophore and xanthophore counts of adult *pomca-/-* and WT control fish in 1mm$^2$ regions after epinephrine treatment. Difference in each category is statistically insignificant (n.s; p-value = 0.07). Data shown as mean ±standard error of the mean (SEM). Axis denote anterior (A) to posterior (P) positioning of the fish. Scale bar: 1 mm (A-F).
(TIF)

**S1 Data. Data used to generate Fig 2D.** Melanosome counts in ROI of black adapted control, *Tg(hs:mch1)* or Tg(hs:mch2) larvae at 7 dpf after heat-shock. The ROI is the area from the posterior of the eyes to the posterior of the hindbrain.
(XLSX)

**S2 Data. Data used to generate Fig 3D and 3G.** Melanosome counts in control or *mchr2* homozygous mutant larvae at 7 dpf when either black adapted or white adapted in ROI from the posterior of the eyes to the posterior of the hindbrain.
(XLSX)

**S3 Data. Data used to generate Fig 4E.** Melanosome counts in black adapted control, *mchr2* homozygous mutant, *mchr2* homozygous mutant; *Tg(hs:mch1)* or *mchr2* homozygous mutant; *Tg(hs:mch2)* larvae at 7 dpf in ROI. Non-significant differences as measured by t-test are labeled as "n.s."
(XLSX)

**S4 Data. Data used to generate Fig 5F and 5K.** Melanosome counts in black adapted control or *pomca* homozygous mutant larvae and in white adapted control, *pomca* homozygous mutant, *mchr2* homozygous mutant or *pomca; mchr2* double homozygous mutant 7 dpf in ROI. Non-significant differences as measured by t-test are labeled as "n.s."
(XLSX)

**S5 Data. Data used to generate S4G Fig.** Dorsal counts of melanophores and xanthophores of adult *pomca-/-* fish compared to counts in WT control fish in 1mm$^2$ regions after epinephrine treatment. Non-significant differences as measured by t-test are labeled as "n.s."
(XLSX)

## Acknowledgments

We thank members of the Mourrain lab for insightful discussions and help. We are grateful to Dr. Gordon Wang for helpful advice for the pigmentation coverage analysis and to Dr. Neil C. Chi for the gift of the *kdrl:mCherry* transgenic line. We would also like to thank the Stanford CSIF Imaging platform.

## Author Contributions

**Conceptualization:** Romain Madelaine, Philippe Mourrain.

**Data curation:** Romain Madelaine.

**Formal analysis:** Romain Madelaine, Keri J. Ngo.

**Funding acquisition:** Philippe Mourrain.

**Investigation:** Romain Madelaine, Keri J. Ngo.

**Methodology:** Romain Madelaine, Keri J. Ngo, Gemini Skariah.

**Supervision:** Philippe Mourrain.

**Validation:** Romain Madelaine, Keri J. Ngo.

**Visualization:** Romain Madelaine, Keri J. Ngo.

**Writing – original draft:** Romain Madelaine, Keri J. Ngo, Philippe Mourrain.

**Writing – review & editing:** Romain Madelaine, Keri J. Ngo, Philippe Mourrain.

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
