## [Decision Letter · Decision Letter 0]

17 Aug 2020

Dear Dr Mourrain,

Thank you very much for submitting your Research Article entitled 'Genetic deciphering of the antagonistic activities of the melanin-concentrating hormone and melanocortin pathways in skin pigmentation' to PLOS Genetics. Your manuscript was fully evaluated at the editorial level and by independent peer reviewers. The reviewers appreciated the attention to an important problem, but raised some substantial concerns about the current manuscript. Based on the reviews, we will not be able to accept this version of the manuscript, but we would be willing to review again a much-revised version. We cannot, of course, promise publication at that time.

In particular you should provide more detail on the pomca mutants as requested by Reviewer 1, and give due consideration to the comments of Reviewer 2 regarding Figure S3 and the abnormal appearance of the the larva in S3H and the different age of the larvae in this Figure. In addition the comments of Reviewer 3 regarding ligand cross-binding and receptor dimerisation should be taken into account. If quantification of tyrosinase and pomca expression, as requested by Reviewer 2, is available it should be included. 

I would also suggest that you reduce the emphasis on the relevance of MCH to human pigmentation. There is no evidence that the MCH system plays any role in mammalian pigmentation. As you know, mice lack MCHR2 completely, and KO of MCHR1 has no pigmentary phenotype.

Please also note that one of the reviewers commented that "data for the graphs in figs 2, 3, 4 and 5 are not provided in spreadsheet form as supporting information" - this must be addressed in your revision in accordance with PLOS Genetics' open access data policy.

If you decide to revise the manuscript for further consideration at PLOS Genetics, please aim to resubmit within the next 60 days, unless it will take extra time to address the concerns of the reviewers, in which case we would appreciate an expected resubmission date by email to plosgenetics@plos.org.

We note that numerical data for Figures 2, 3, 4 and 5 should be included as supplementary data.

[LINK]

We are sorry that we cannot be more positive about your manuscript at this stage. Please do not hesitate to contact us if you have any concerns or questions.

Yours sincerely,

Ian J. Jackson

Guest Editor

PLOS Genetics

Gregory P. Copenhaver

Editor-in-Chief

PLOS Genetics

Reviewer's Responses to Questions

**Comments to the Authors:**

Reviewer #1: In the manuscript entitled ’Genetic deciphering of the antagonistic activities of the melanin-concentrating hormone and melanocortin pathways in skin pigmentation’ Madelaine et al. show that background adaptation in zebrafish, i.e. aggregation or dispersal of melanosomes within melanophores, is regulated by two opposing signaling systems. The authors show that projections from melanin-concentrating hormone (MCH) producing neurons are found in close proximity to blood vessels in the pituitary, suggesting that this is the way the peptides are released to exert their functions in the periphery. Overexpression of the two MCH orthologs induces the aggregation of melanosomes in larval zebrafish, and the authors show that the peptide hormones act specifically via one of the receptors, Mchr2. Mutations in the gene encoding this receptor affect background adaptation not only in larvae but also in adult fish. In addition they show that the opposite process, i.e. melanosome dispersal, is defective in larvae with mutations in the pomca gene, which presumably affects melanocortin signaling via the receptor Mc1r. Surprisingly, while mchr2-/- is epistatic over pomca-/- in larvae, this is no longer the case in adult fish, which hints at a more complex regulation of the process.

The manuscript is generally well written and easy to understand; I only think that the introduction and the discussion could be a little bit more concise.

The results are presented in a clear fashion and the conclusions are sound.

The only point, which I feel should be clarified concerns the pomca mutants generated and the relation to Mc1r signaling:

- the authors describe that two small deletions in different exons were produced, but they fail to indicate which allele (or combination) the animals shown in figures 4 and 5 actually carry.

- are the phenotypes of animals homozygous for either mutant allele identical?

- how does the pomca k.o. phenotype compare to the published mc1r k.o. phenotype?

- especially, what is the number and distribution of pigment cells in the adult mutant fish? Is it similar to wild type, meaning only background adaptation in the mutant larvae is affected and there is no adult phenotype? Or, is it similar to mc1r mutants, which have not been described to be defective in background adaptation, but rather in counter shading? This could possibly provide further clues to understanding the complexity of the system.

Reviewer #2: PGENETICS-D-20-01069

'Genetic deciphering of the antagonistic activities of the melanin-concentrating hormone and melanocortin pathways in skin pigmentation'

The authors use loss of function mutants for mchr1a/b, mchr2 and pomca, plus transgenic reporters of MCH peptide expression, to test the hypothesis that antagonistic interactions between MCH and MSH signalling control background adaptation (i.e. the rapid movement of melanosomes along the microtubular cytoskeleton to change the overall darkness of the body) in a fish genetic model, zebrafish.

The authors begin by generating a transgenic reporter of mch2 expression in zebrafish, revealing projection of MCH2-expressing hypothalamic neurons to the pituitary, and specifically their projection to regions in the vicinity of the associated vasculature. As presented the relative positions of the areas shown in each panel are difficult for the non-expert to interpret. An interpretative diagram of brain structure and the zones shown would be invaluable.

They then used 2 distinct transgenic lines to drive heat shock-mediated overexpression of MCH1 and MCH2 respectively, compared to control non-transgenic fish that were also heat shocked. Each peptide was capable of reversing melanosome dispersion of early larval fish kept on a dark background, when expressed by heat shock.

To assess which of the candidate MCHR might mediate the MCH effect, the authors generated mchr1a mchr1b and mchr2 knockout lines, generated by CRISPR/Cas-9 or TILLING methodologies. Both 7 dpf and adult mchr2 knockouts show a pronounced constitutive dispersion of melanin, under both light and dark background conditions, as expected of the background adaptation-associated receptor. As expected, this effect was not affected by over-expression of mch1 or mch2. In contrast, neither single nor double mutants of mchr1 genes gave a pigment dispersion phenotype.

As a prelude to assessing the endogenous role of POMCA in melanosome dispersion, the authors assess tyrosinase expression in melanocytes as a method for evaluating whether melanocyte differentiation is normal. However, this requires a quantitative assessment and whole mount ISH technique they use is, at best, seim-quantitative. The authors should supplement this panel with qRT-PCR quantitation of tyrosinase expression in the body (i.e a region excluding the eyes). Likewise, they assess pomca expression in mchr2 mutants by WISH, and by immunofluorescence, to conclude that MCHR2 knockout does not affect pomca or MSH peptide levels. Again, since they are drawing quantitative conclusions from inherently semi-quantitative techniques, more sensitive quantitative data would be useful to supplement these observations.

To test the hypothesis that melanosome dispersion depends upon POMC/a-MSH, the authors used transgenic overexpression of POMCA. Their demonstration of enhanced dispersion (Fig. S3G,H) is unclear. The larva shown in panel H seems to be deformed, with abnormal distribution of melanocytes. This makes it very difficult to assess the dispersion of melanosomes. Closeups of individual melanocytes in morphologically unaffected regions, and quantitation of the degree of dispersion (as used throughout the rest of the paper) should be provided to justify the conclusion reached. Also, the fish shown are much younger than in the rest of the study (3 dpf) and hence close to the time when background adaptation first appears. Hence, comparison of control and transgenic fish on both white and dark backgrounds should be used to control for this.

To more directly test the role for POMC, the authors generated a pomca mutant. They demonstrate both the absence of immunofluorescently-detectable aMSH and the expected failure of melanosome dispersion under light-adapted conditions. Intriguingly, an mchr2; pomca double knockout shows a 7 dpf larval phenotype indistinguishable from that of mchr2 alone, indicating that factors other than aMSH can promote melanosome dispersion. The authors observe that this changes gradually as the fish age, so that adult double mutants show the pomca knockout phenotype, indicating that another factor can substitute for MCH at this stage. These observations are intriguing and setup a series of future studies as outlined in the Discussion.

In summary, this is a conceptually simple, but convincing, well-controlled and nicely documented demonstration of the specific MCH signalling components underpinning zebrafish background adaptation.

Minor points

1) Fig. 1 legend. Outline in panel A needs explanation.

2) Fig. 2D. x axis label font is too small. Enlarge so is readily legible.

3) Fig. 3 legend. ‘ROI’ is not defined, although presumably ‘Region of Interest’. Also, this region should be indicated/described briefly.

4) Fig. 5 title. As written, it does not quite make sense to me. Authors should revise.

Reviewer #3: In this study, Romaine and colleagues investigate the genetic mechanisms underlying skin color in teleosts (zebrafish) using transgenic models and pigmentation analysis.

These new original genetic evidence for an antagonistic role of MCH and alpha-MSH in skin pigmentation are convincing and confirm old antibody-based studies. The manuscript is well written and data are rigorously presented and analysed. I did appreciate some of the control experiment (e.g., double transgenic). The discussion of the results is appropriate. Thus, I have several comments and questions that are listed below in order of appearance in the MS.

The authors first map the axonal projections of MCH neurons using a transgenic line Tg(mch2:egfp) which nicely confirm original immunostaining study in salmon from the '80s. Further they show that both MCH1 and MCH 2 share pigment aggregation properties. At this point, I was wondering if MCH1 and MCH2-expressing neurons project to other targets outside the pineal gland with some possible distinction amongst targets - this should be mentioned somewhere in the MS, at least for MCH2-expressing cells, and if not reported previously.

To study the receptor subtypes involved in this regulation, they then generate mchr1a, mchr1b and mchr2 KO mutants (CRISPR/Cas or TILLING) and found that only mchr2 is involved in skin pigmentation. I did appreciate the careful analysis of the double mutant [mchr1a-mchr1b] controls which did not show neither changes in pigmentation. Interestingly, control experiments using mch1 or mch2 over-expression in the mchr2 mutant confirmed the need of a functional MCHR2 to reduce melanin dispersion. This is a very convincing result.

Based on previous studies showing an antagonistic action between MCH and alpha-MSH, the authors generate a KO of pomca (POMC precursor gene) fish that is not able to adapt to a black background. Yet, knock-out of pomca in the mchr2 mutant does not rescue the mchr2 and double mutant phenotype were different than the larval stage. The authors interpret these data as "... additional factors other than α-MSH can promote melanosome dispersion in the mchr2 mutant genetic background in larvae" and "...the inability of the adult mchr2 mutant to adapt to a white background is due to the predominant activity of the melanocortin pathway".

This is puzzling and I was wondering whether this could be to a low affinity cross-binding between peptide ligand (MCH/alpha-MSH - e.g., MSH onto MCHR2) and the remaining receptors. Furthermore, are the MCH and alpha-MSH peptides known to bind a single receptor (MCH or MSH receptors subtypes) with low affinity ? Could that explain this phenotype ?

Homodimerization and heterodimerization are frequent amongst GPCRs - since those receptors are co-expressed in skin melanophores, could such phenomenon affect the intracellular pathway activation and, ultimately the skin pigmentation in the larval stage of these mutants?

Does a change in gene transcripts, peptide or receptor expression may explain the larval versus adult different phenotypes ? In other words, are there known changes of these across development including 'peptide-receptor' affinity ? If so, could the authors provide such experimental data ? that would really strengthen the study.

Figure 2A, 3 A,B, 5 G,H all show control conditions, however, the pigmentation shape is quite different - where does that come from ? is this due to the background exposure? is there any type of correction for that across quantifications ?

Minors:

-many results section start with a rationale that often repeat some parts of the introduction - repetition should be avoided.

-avoid the use the form "to our knowledge" multiple times.

**Have all data underlying the figures and results presented in the manuscript been provided?**

Reviewer #1: **No: **data for the graphs in figs 2, 3, 4 and 5 are not provided in spreadsheet form as supporting information

Reviewer #2: Yes

Reviewer #3: Yes

PLOS authors have the option to publish the peer review history of their article (what does this mean?). If published, this will include your full peer review and any attached files.

Reviewer #1: **Yes: **Uwe Irion

Reviewer #2: No

Reviewer #3: **Yes: **Antoine Adamantidis

---

## [Decision Letter · Decision Letter 1]

30 Oct 2020

Dear Dr Mourrain,

We are pleased to inform you that your manuscript entitled "Genetic deciphering of the antagonistic activities of the melanin-concentrating hormone and melanocortin pathways in skin pigmentation" has been editorially accepted for publication in PLOS Genetics. Congratulations!

Could you please note the two minor comments from Reviewer 2, below, and make appropriate changes to the manuscript as you prepare your final draft for the production team (the editorial team will not need to reevaluate).

Yours sincerely,

Ian J. Jackson

Guest Editor

PLOS Genetics

Gregory P. Copenhaver

Editor-in-Chief

PLOS Genetics

Comments from the reviewers (if applicable):

Reviewer's Responses to Questions

**Comments to the Authors:**

Reviewer #1: The authors adequately addressed the points I raised previously; the manuscript should be published.

Reviewer #2: 'Genetic deciphering of the antagonistic activities of the melanin-concentrating hormone and melanocortin pathways in skin pigmentation'

The authors have dealt appropriately with the referees’ comments. I have only a couple of minor suggestions:

Fig. 2D and Fig. 3D legends. Authors refer to ‘bottom’ of eyes and hindbrain, but more correct would be ‘posterior’. However, I wonder if the region might be best described as ‘those melanocytes in the anterior Dorsal Stripe lying dorsal to the hindbrain’?

p. 11 ‘premature apparition of a stop codon’ is used; better would be ‘appearance of a premature stop codon’

Reviewer #3: The authors have satisfactorily answered all my concerns.

**Have all data underlying the figures and results presented in the manuscript been provided?**

Reviewer #1: Yes

Reviewer #2: Yes

Reviewer #3: Yes

PLOS authors have the option to publish the peer review history of their article (what does this mean?). If published, this will include your full peer review and any attached files.

Reviewer #1: **Yes: **Uwe Irion

Reviewer #2: **Yes: **Robert N. Kelsh

Reviewer #3: **Yes: **Antoine Adamantidis

**Data Deposition**

http://datadryad.org/submit?journalID=pgenetics&manu=PGENETICS-D-20-01069R1

**Press Queries**

---

## [Editor Report · Acceptance letter]

1 Dec 2020

PGENETICS-D-20-01069R1 

Genetic deciphering of the antagonistic activities of the melanin-concentrating hormone and melanocortin pathways in skin pigmentation 

Dear Dr Mourrain, 

We are pleased to inform you that your manuscript entitled "Genetic deciphering of the antagonistic activities of the melanin-concentrating hormone and melanocortin pathways in skin pigmentation" has been formally accepted for publication in PLOS Genetics! Your manuscript is now with our production department and you will be notified of the publication date in due course.

With kind regards,

Nicola Davies

PLOS Genetics

On behalf of:
